# Pericardial NT-Pro-BNP and GDF-15 as Biomarkers of Atrial Fibrillation and Atrial Matrix Remodeling in Aortic Stenosis

**DOI:** 10.3390/diagnostics11081422

**Published:** 2021-08-05

**Authors:** Mariana Fragão-Marques, Isaac Barroso, Rui Farinha, Isabel M Miranda, Diana Martins, Jennifer Mancio, João Rocha-Neves, João T Guimarães, Adelino Leite-Moreira, Inês Falcão-Pires

**Affiliations:** 1Cardiovascular Research and Development Center, Faculty of Medicine of the University of Porto, 4200 Porto, Portugal; imiranda@med.up.pt (I.M.M.); dlfm94@gmail.com (D.M.); jennimancio@gmail.com (J.M.); afleitemoreira@gmail.com (A.L.-M.); pires@med.up.pt (I.F.-P.); 2Department of Clinical Pathology, São João University Hospital Centre, 4200 Porto, Portugal; isaacbarroso@gmail.com (I.B.); rmrbfarinha@sapo.pt (R.F.); jtguimar@med.up.pt (J.T.G.); 3EPIUnit, Instituto de Saúde Pública, University of Porto, 4200 Porto, Portugal; 4Department of Anatomy, Faculty of Medicine, University of Porto, 4200 Porto, Portugal; joaorochaneves@hotmail.com

**Keywords:** atrial fibrillation, aortic stenosis, cardiac surgery, pericardial fluid, biomarker, GDF-15, NT-pro-BNP

## Abstract

Aims: This study aimed to evaluate the association of GDF-15 and NT-pro-BNP in two different biological matrices with AF in severe aortic stenosis patients undergoing aortic valve replacement surgery (AVR), its association with atrial matrix remodeling, as well as with 30-day postoperative outcomes. Main Methods: One hundred and twenty-six patients between 2009 and 2019 with severe aortic stenosis undergoing AVR surgery in a tertiary hospital were assessed. Key Findings: pericardial fluid GDF-15 and pericardial fluid and serum NT-pro-BNP were increased in AF patients with aortic stenosis. COL1A1 and COL3A1 gene expression increased when pericardial fluid NT-pro-BNP values were higher. TIMP4 was positively correlated with pericardial fluid GDF-15. Significance: GDF-15 and NT-pro-BNP in the pericardial fluid are biomarkers of atrial fibrillation in aortic stenosis and correlate with atrial matrix remodeling. AKI is predicted by both serum and pericardial fluid GDF-15.

## 1. Introduction

Atrial fibrillation (AF) is the most common sustained arrhythmia with adverse clinical outcomes [1]. It is estimated that AF affects 2% to 4% of the general population, having increased its prevalence over time [2]. Although some mechanisms of AF pathophysiology are widely recognized, namely, rapid ectopic firing, early and delayed after-depolarisations, rotors and electrical and structural remodeling, individual risk factors for AF are associated with specific pathophysiological mechanisms [3]. Aortic stenosis induces pressure overload in left cardiac chambers and shares multiple risk factors with AF, namely, age and hypertension. Thus, the prevalence of pre-existing AF in aortic stenosis patients undergoing aortic valve replacement surgery (AVR) ranges from 8% to 13% [4]. Specific mechanisms of AF in this subset of patients are still unclear, although age and hypertension have been associated with structural remodeling [3].

Growth Differentiation Factor 15 (GDF-15) is a stress-responsive member of the transforming growth factor-β superfamily, implicated in fibrotic processes [5]. It has been associated with bleeding risk in atrial fibrillation (AF) [5], in addition to the onset of AF in population-based cohorts [6], as well as AF following coronary artery bypass surgery [7,8]. However, results are inconsistent and non-existing in aortic stenosis.

Moreover, other biomarkers, such as NT-pro-B-type natriuretic peptide (NT-pro-BNP), have been consistently associated with atrial fibrillation [6,9]. NT-pro-BNP predicts mortality in valvular disease, including aortic stenosis, although the role in atrial fibrillation in this subset of patients remains unclear [10].

Despite serum being the most common biological matrix in which these biomarkers are tested, pericardial fluid concentrates plenty of heart-derived factors and it is easily obtained in the setting of cardiac surgery or pericardiocentesis [11]. Recently, pericardial fluid T reg cells and NT-pro-BNP were increased in chronic heart failure and C-type Natriuretic Peptide levels were higher in patients with lower ejection fraction undergoing coronary artery bypass surgery [12,13]. Nevertheless, it remains underrated and poorly studied, particularly in severe aortic stenosis undergoing AVR.

This study aimed to evaluate the association of GDF-15 and NT-pro-BNP in two different biological matrices (serum and pericardial fluid) with AF in severe aortic stenosis patients undergoing AVR, its association with atrial matrix remodeling, and 30-day postoperative outcomes.

## 2. Materials and Methods

### 2.1. Study Design

A post hoc analysis of a prospective study on a cohort of aortic stenosis patients was made. Between 2009 and 2019, 136 patients with severe aortic stenosis undergoing AVR surgery in a tertiary hospital were consecutively recruited. Patients were 18 years of age and older, with all sexes eligible for the study. Participants had symptomatic severe aortic stenosis (defined as aortic valve area of <1 cm^2^ or 0.6 cm^2^/m^2^ by transthoracic echocardiography) referred to aortic valve replacement. Exclusion criteria included coexisting severe aortic valve regurgitation, reoperations, emergent cases and active malignancy.

Overall, baseline clinical and echocardiography data from 126 patients were documented and 65 pericardial fluid samples, 95 serum samples and 36 atrial samples were collected during and immediately before surgery. Recruited patients were then grouped according to their cardiac rhythm status: AF or Sinus Rhythm (SR) (see below).

Sample size was calculated for differences between AF and SR patients concerning pericardial fluid and serum GDF-15 and NT-pro-BNP, according to literature-based means and standard deviations between groups, with an allocation ratio of approximately 0.25, a power of 80% and a probability of a type I error of 5%. Sample size was based on the highest needed number of samples between biomarkers (GDF-15 and NT-pro-BNP) and biological matrices (serum and pericardial fluid), with pericardial fluid GDF-15 quantification requiring a total of 79 patients (63 SR, 16 AF), considering a difference between patient subgroups of 120 ng/L and a standard deviation of 150 ng/L.

This study follows the Declaration of Helsinki. The protocol was approved by the institution’s ethics committee (Comissão de Ética para a Saúde (CES) do Centro Hospitalar de São João/Faculdade de Medicina da Universidade do Porto) and data confidentiality was assured. All participants gave written informed consent.

### 2.2. Data Collection

Patients were classified as smokers when reporting active smoking or smoking cessation during the past year. Chronic Lung Disease (CLD) was considered when a diagnosis of Chronic Obstructive Pulmonary Disease, Obstructive Sleep Apnea or Interstitial Lung Disease was present, or if there was a significant decrease in lung function in a pre-operative spirometry. When there was a previous myocardial infarction or invasive coronary angiography reporting stenosis >50%, Coronary Artery Disease (CAD) was reported. Additionally, cerebrovascular disease was defined based on a history of stroke or transitory ischemic attack or a reported stenosis >50% on carotid Doppler ultrasonography. Symptoms of heart failure were classified under the New York Heart Association (NYHA) functional classification [14]. Basal Creatinine was measured at admission through *OLYMPUS 5400/5800* chemistry analyzer from Beckman Coulter^®^ (Brea, California, United States) using a modified Jaffe Method and basal hemoglobin was measured through the sodium lauryl sulphate method in the *Sysmex XE-2100* from Sysmex Corporation^®^ (Kobe, Hyogo, Japan). Hemoglobin variation was calculated as the difference between discharge and admission values, and a dichotomous variable was created using the 75th percentile.

Moreover, echocardiography was performed up to 6 months prior to surgery by experienced operators. Cardiac chamber dimensions and volumes were measured as recommended and systolic function was accessed by evaluation of left ventricular ejection fraction, using the modified Simpson rule from biplane 4 and 2 chamber views [15]. Reduced ejection fraction was considered below 40%, according to the European Society of Cardiology guidelines [16].

### 2.3. Endpoints

All patients had an electrocardiogram (EKG) performed up to 6 months prior to surgery. AF was defined according to international guidelines as absolutely irregular RR intervals and no discernible, distinct P waves, with an episode lasting at least 30 s being the threshold for diagnostic purposes [17]. AF was further defined by previous clinical records and EKGs (more than 7 days) to exclude paroxysmal AF; patients were continuously monitored during surgery and up to 48 h afterwards by telemetry during hospital stay, with daily annotations of irregular cardiac rhythm thereafter by the attending physician. Patients were then divided into SR and AF groups, as previously mentioned.

Outcomes at 30 days were recorded, namely, MACE (stroke, acute myocardial infarction, all-cause mortality and major arrhythmia–ventricular fibrillation and complete atrioventricular block); acute kidney injury (AKI), defined by an increase in 0.3 mg/dL in creatinine; and postoperative atrial fibrillation (POAF), which was monitored by telemetry during the first 48 h post-surgery and afterwards by EKG if an arrhythmic pulse was detected [18]. Increased length of stay (LOS) was defined as a dichotomous variable using the 75th percentile.

### 2.4. Sample Collection and Preparation

As previously mentioned, 65 pericardial fluid, 95 serum and 36 atrial samples (29 for histology assessment and 19 for gene expression analysis) were gathered. Whole blood samples were collected from a peripheral vein in tubes with clot activator and separation gel immediately before surgery, and centrifuged at 870× *g* for 10 min; the resulting serum was stored at −80 °C. Pericardial fluid samples were collected at the time of pericardial section during surgery, and centrifuged at 870× *g* for 10 min and frozen at −80 °C. During surgery, an atrium appendix myocardial biopsy was collected, and the sample was either immediately fixed in formalin (formalin solution, neutral buffered, 10%) and processed for histological analysis or flash-frozen in liquid nitrogen and stored at −80 °C for posterior gene expression analysis.

### 2.5. GDF-15 and NT-Pro-BNP

GDF-15 and NT-pro-BNP in serum (*n* = 95) and pericardial fluid (*n* = 65) were both quantified by immunoassay on *COBAS e411* from Roche^®^ (Basel, Switzerland).

### 2.6. Histomorphometric Quantification

All samples were processed using a *Histocore Pearl* automatic processor from Leica^®^ (Wetzlar, Germany). A total of 29 patients (SR *n* = 19, AF *n* = 10) were used to assess atrial fibrosis. Paraffin inclusion was performed in a *Histocore Arcardia Embedding System* from Leica^®^ (Wetzlar, Germany). Fibrosis was assessed using 3 µm atrial sections stained with Red Sirius and quantification was performed with *Image-Pro Plus* from Media Cybernetics Inc^®^ (Rockville, Maryland, United States) in 8 fields per sample using the 100× amplification (results presented as percentage of the myocardial area of each section).

### 2.7. Gene Expression Analysis

A total of 19 patients (SR *n* = 11, AF *n* = 8) were evaluated for the gene expression of several extracellular matrix proteins: collagen I (COL1A1) and III (COL3A1), Tissue Inhibitor of Metalloproteinase (TIMP) 1, TIMP4, Matrix Metalloproteinase (MMP) 2 and MMP9. RNA extraction was performed using a TRIzol protocol following the manufacturer’s instructions and RNA concentration and integrity were assessed both in the *Nanodrop* from Thermo Fisher Scientific^®^ (Waltham, Massachusetts, United States) and by electrophoresis. cDNA (100 ng/µL) was synthesized using the *SensiFAST^TM^ cDNA Synthesis Kit* from Meridian Bioscience Inc^®^ (Cincinnati, Ohio, United States), with reactions conducted in a *T100 thermal cycler* from Bio-Rad^®^ (Hercules, California, United States). RT-qPCR reactions were performed using the *PikoReal^TM^ Real-Time PCR System* from Thermo Fisher Scientific^®^ (Waltham, Massachusetts, United States), using previously described protocols [19]. Before gene expression quantification using the 2^−ΔCT^ method, PCR efficiency of each gene, including the internal control gene (18s RNA) was determined and it was assured they were identical.

Target gene expression was normalized to the 18S gene. Initial mRNA expression data were logarithm transformed using the 2^−ΔCT^ method [20].

### 2.8. Statistical Analysis

Categorical variables were presented as percentages and continuous variables as mean and standard deviation (sd) or median and quartile 1 and quartile 3 (Q1 and Q3), according to normality testing. Chi-squared test or Fisher’s exact test were used to analyze categorical variables when appropriate and continuous variables were compared with a *t*-test for independent samples or with Mann–Whitney U test, according to data normality. Correlations between matrices of target biomarkers were performed and the Pearson’s correlation coefficient was presented as the measure of association. Linear regression was used to test for correlations between biomarkers and extracellular matrix gene expression.

Statistical significance was considered when *p* < 0.05. *SPSS Statistics version 25* from IBM^®^ (Armonk, New York, NY, USA) was used for all statistical analyses.

## 3. Results

### 3.1. Patient Demographics

Recruited patients were divided into two groups, according to pre-operative baseline rhythm: sinus rhythm (*n* = 107) or atrial fibrillation (*n* = 19). Patients had a mean age of 72.2 ± 8.4 years (mean ± sd), of which 48.4% (*n* = 61) were male. Hypertension was the most common comorbidity, representing 75.4% (*n* = 95) of patients, followed by dyslipidemia, diabetes mellitus and chronic lung disease. Symptoms of heart failure (NYHA ≥ II) were present in 64.3% (*n* = 81) of patients (Table 1).

Concerning differences between AF and SR groups, no statistically significant differences were demonstrated in baseline patient characteristics. AF patients were 74.0 ± 9.1 years vs. 71.9 ± 8.3 years in SR patients (*p* = 0.306), while males represented 42.1% of AF and 49.5% of SR participants (*p* = 0.551). In addition, hypertension was the most common comorbidity in both subgroups, with 72.2% vs. 78.8% in AF and SR patients (*p* = 0.532). Symptoms of heart failure were present in 75.0% vs. 78.4% of subjects, respectively (*p* = 0.749) (Table 1).

### 3.2. Baseline Echocardiographic Assessment

Patients had an overall preserved ejection fraction and dilated left atrium. Gradients represented severe aortic stenosis, with an Aortic Valve Mean Gradient (AMeanG) of 50.4 ± 12.6 mmHg and aortic valve area (AVA) of 0.8 ± 0.2 cm^2^ (Table 1).

Echocardiography performed before surgery demonstrated AF patients had an increase in LAD when compared with their SR counterparts (49.4 ± 7.2 mm vs. 41.0 ± 9.6 mm, *p* = 0.038). The remaining echocardiographic parameters were similar between groups and are depicted in Table 1.

### 3.3. Pericardial Fluid and Serum GDF-15 and NT-Pro-BNP

#### 3.3.1. Atrial Fibrillation

Results are presented in Figure 1. GDF-15 was only increased in AF patients with aortic stenosis when measured in the pericardial fluid (median (Q1–Q3) 775.6 (660.0–1274.8) ng/L vs. 616.2 (401.7–983.9) ng/L, *p* = 0.031); in fact, serologic GDF-15 remained similar between groups (1697.0 (1040.5–2727.5) ng/L vs. 1102.0 (764.7–1835.8) ng/L, *p* = 0.124). Concerning NT-pro-BNP, this biomarker was increased in AF patients in both biological matrices–6843.0 (1522.8–10830.3) ng/L vs. 952.6 (501.6–1997.5) ng/L, *p* < 0.001 in pericardial fluid; and 827.2 (668.1–1495.0) ng/L vs. 272.7 (146.2–601.3) ng/L, *p* = 0.002 in serum. When testing for the diagnostic accuracy of significantly altered biomarkers, AUROC (Area Under the Receiver Operating Characteristics) was measured, with pericardial fluid NT-pro-BNP having the highest area (0.9; 95% Confidence Interval (CI) [0.8–1.0]), followed by serum NT-pro-BNP (0.8; [0.7–0.9]) and pericardial fluid GDF-15 (0.7; [0.6–0.8])—Figure 2.

#### 3.3.2. Atrial Matrix Remodeling

Atrial remodeling was tested through the quantification of tissue fibrosis and target expression of extracellular matrix genes. The biomarkers had a lognormal distribution; thus, a logarithm transformation was used. Data are depicted in Table 2.

Fibrosis was similar regardless of biomarker quantification. Furthermore, COL1A1 and COL3A1 gene expression increased when pericardial fluid NT-pro-BNP values were higher (*p* = 0.005 and *p* = 0.033, respectively), while serum NT-pro-BNP had no correlation with either collagen gene. (Figure 3) On the other hand, neither pericardial fluid nor serum GDF-15 correlated with COL1A1 and COL3A1 expression.

Specific remodeling genes were tested, with TIMP4 being positively correlated with pericardial fluid GDF-15 (*p* = 0.025), although the same result was not verified in serum. Either serum or pericardial fluid NT-pro-BNP were similar irrespective of TIMP4 gene expression. Conversely, TIMP1 and MMP9 were comparable regardless of the matrix or biomarker. Additionally, MMP2 expression was higher when pericardial fluid NT-pro-BNP increased (*p* = 0.023), even though no differences were observed in serum. GDF-15 in serum and pericardial fluid remained identical regardless of MMP2 gene expression (Figure 3).

### 3.4. Thirty-Day Postoperative Outcomes

#### 3.4.1. Atrial Fibrillation

Postoperative outcomes were assessed during the first 30 days post-surgery, demonstrating an overall MACE prevalence of 19% (*n* = 24). POAF (Postoperative Atrial Fibrillation) occurred in 27.8% (*n* = 35) of cases, while AKI affected patients in a proportion of 27.0% (*n* = 34). Hemoglobin variation was superior to the 75th percentile in 23.8% (*n* = 30) of participants and LOS was prolonged (>75th percentile) in 22.2% (*n* = 28) of cases.

Differences in 30-day outcomes between AF and SR subgroups demonstrated an increase in postoperative AKI (*p* = 0.001) and increased bleeding (Hb variation, *p* = 0.017). Data are shown in Table 3.

#### 3.4.2. Pericardial Fluid and Serum GDF-15 and NT-Pro-BNP

Pericardial fluid GDF-15 was significantly increased in AKI (*p* = 0.027), although similar regardless of MACE, POAF occurrence, postoperative bleeding or LOS.

Likewise, serum GDF-15 was increased when patients presented with AKI in the postoperative period (*p* = 0.015). There were no differences in serum GDF-15 values concerning MACE or bleeding. Patients who developed POAF had higher serum GDF-15, demonstrating a tendency towards significance (*p* = 0.076). Similarly, increased LOS was associated with higher GDF-15 serum quantifications, nearing statistical significance (*p* = 0.085).

Pericardial fluid NT-pro-BNP was higher in AKI, although non-significant (*p* = 0.096). Additionally, NT-pro-BNP was similar regardless of MACE, POAF occurrence, bleeding or hospital stay.

Moreover, NT-pro-BNP in serum was increased in patients with POAF, demonstrating a trend towards significance (*p* = 0.096). Serum NT-pro-BNP values were similar independently of MACE, AKI, bleeding or LOS (Table 4, Figure 4).

### 3.5. Comparison between Pericardial Fluid and Serum Measurements

The biomarkers presented a lognormal distribution; thus, a logarithm transformation was used. GDF-15 blood measurements were significantly higher when compared with pericardial fluid quantifications (median (Q1–Q3) 1162.0 (769.7–1838.0) ng/L vs. 703.5 (437.7–1026.3) ng/L, *p* < 0.001, respectively). Conversely, NT-pro-BNP values were lower in serum than in pericardial fluid (345.4 (157.7–798.7) ng/L vs. 1201.5 (631.6–3892.5) ng/L, *p* < 0.001). Correlations between serum biomarkers and their pericardial fluid counterparts were made, demonstrating a good correlation for NT-pro-BNP (r = 0.869, *p* < 0.0001) and a fair correlation (r = 0.736, *p* < 0.0001) for GDF-15 (Figure 5).

## 4. Discussion

Patients with AF had a higher left atrium diameter and increased pericardial fluid GDF-15; both serum and pericardial fluid NT-pro-BNP were increased in these patients. The diagnostic accuracy for AF was superior with pericardial fluid NT-pro-BNP. Increased NT-pro-BNP in the pericardial fluid was associated with higher atrial expression of collagen type I and type III and MMP2 extracellular matrix genes, while higher pericardial fluid GDF-15 quantifications was associated with higher TIMP4 atrial gene expression. Additionally, AF patients had a higher prevalence of postoperative AKI and bleeding, and either serum or pericardial fluid GDF-15 predicted the former outcome. GDF-15 had increased serum values when compared with pericardial fluid and NT-pro-BNP had higher pericardial fluid concentrations relative to serum matrix. Both biomarkers presented a correlation between the two biofluids.

### 4.1. GDF-15 in Atrial Fibrillation

Serum GDF-15 has been consistently identified as a predictor of stroke, mortality and bleeding in atrial fibrillation, either alone or included in an ABC (Age, Biomarkers, Clinical history) risk score [5,21]. Chan et al. observed GDF-15 in both heart failure with preserved ejection fraction and reduced ejection fraction was independently associated with atrial fibrillation, although Santema et al. did not find a difference between AF and SR in similar patients [22,23]. Furthermore, in a cohort of the Framingham Heart Study, incident AF was not independently predicted by GDF-15 or ST2 biomarkers [24]. Although a study in hypertrophic cardiomyopathy reported an association of serum GDF-15 with AF diagnosis, this has not yet been assessed in aortic stenosis cohorts. The present manuscript describes similar quantifications in aortic stenosis patients with AF or SR patients when considering serum GDF-15 in the AVR setting. Nevertheless, age has been reported to increase GDF-15 serum levels [25], although evidence is scarce regarding the biomarker. In this cohort, age was positively correlated with GDF-15 (*p* = 0.006, data not shown), even though AF and SR patients had no differences in age. Although it could be a potential explanation for the association between GDF-15 and AF, because age is similar regardless of patient cardiac rhythm status in this cohort, the findings appear not to be confounded, although larger cohorts might be necessary to confirm the association. Moreover, AF and heart failure are often associated with an altered heart rate. GDF-15 has recently been reported not to change with heart rate [26], in spite of its potential relevance in connecting heart failure outcomes and mechanisms through GDF-15 quantification.

### 4.2. GDF-15 in Pericardial Fluid

Interestingly, GDF-15 in pericardial fluid had higher values in the AF patient subgroup. Yuan et al. suggested an association between cardiac and kidney function in patients with coronary artery disease and pericardial fluid GDF-15, which was the first study concerning patients with heart disease and the quantification of this biomarker in pericardial fluid [27]. GDF-15 is a stress-responsive member of the transforming growth factor-β superfamily associated with fibrotic processes and inflammation, which could explain its relationship with cardiovascular disease and outcomes; namely, atrial fibrillation [5].

### 4.3. GDF-15 and Extracellular Matrix Remodeling

The present work demonstrates a positive correlation between pericardial fluid GDF-15 and atrial TIMP4, with no correlation with the corresponding serum biomarker. Although there are no studies associating TIMP4 with GDF-15; this extracellular matrix protein has been associated with atrial fibrillation, which could help understand the relationship between GDF-15 and this arrhythmia. Nevertheless, evidence is conflicting, with studies in aortic stenosis being non-existent [28,29]. Remarkably, our results suggest pericardial fluid is more accurate than its serum counterpart in predicting cardiac disease. Several studies have reported an association between serum GDF-15 and fibrosis, as well as MMP2, MMP9 and TIMP1, although in an array of distinct clinical conditions, which was not confirmed in the present manuscript [30,31]. Moreover, GDF15 is widely expressed in multiple tissues. Although serum levels were higher than pericardial fluid levels, it is possible that only GDF15 that is heart-derived is associated with cardiovascular disease prediction. Therefore, serum levels could be less specific, with the pericardial fluid presenting information on cardiac tissue without noise from other peripheral sources.

### 4.4. NT-Pro-BNP in Atrial Fibrillation

NT-pro-BNP is a precursor of BNP and it is synthesized by ventricular myocytes in response to stretch. This biomarker has been reported to increase in AF, although studies in aortic stenosis are lacking [32]. Liu et al. found that AF in hypertrophic cardiomyopathy was associated with higher values of NT-pro-BNP in serum [9]. Herein, this biomarker was increased when AF was present, both in serum and pericardial fluid, with the latter having a more robust association.

### 4.5. NT-Pro-BNP in Pericardial Fluid and Atrial Matrix Remodeling

Michaud et al. measured pericardial fluid NT-pro-BNP in a postmortem setting and found an association with cardiac ischemia, although there are no studies in AF [33]. In accordance with the GDF-15 results, it appears pericardial fluid is more accurate than serum in predicting cardiac diseases. Our results suggest a positive correlation between pericardial fluid NT-pro-BNP and collagen type I, collagen type II and MMP2 atrial gene expression, though Tziakas et al. found that patients with acute myocardial infarction had a negative association between serum NT-pro-BNP and MMP2 and Cao et al. did not find a correlation between serum pro-BNP and collagen type I and III atrial gene expression in valvular atrial fibrillation [34,35]. Our results help understand the relationship between NT-pro-BNP and AF, since atrial remodeling has long been associated with this arrhythmia [36].

Moreover, NT-pro-BNP has been recently associated with outcomes in atrial fibrillation, namely, stroke and all-cause mortality [37]. Considering that pericardial fluid NT-pro-BNP had a better diagnostic accuracy for atrial fibrillation, and that it was associated with atrial remodeling, contrary to the serum values, quantifying NT-pro-BNP in the pericardial fluid at the time of surgery could more precisely identify atrial fibrillation patients at high risk of adverse outcomes. Secondly, as these were associated with non-paroxysmal atrial fibrillation, increased concentrations of serum NT-pro-BNP could help identify patients with higher risk of progression to a more persistent form of the arrhythmia, in addition to atrial arrhythmia burden detected by EKG.

### 4.6. GDF-15 and Acute Kidney Injury

Acute kidney injury has recently been associated with increased levels of baseline serum GDF-15 after coronary artery bypass surgery and cardiac surgery in general [38,39]. There are no studies either regarding acute renal dysfunction following AVR surgery in patients with severe aortic stenosis, or in pericardial fluid. The present work suggests both serum and pericardial fluid GDF-15 as biomarkers of AKI in this subset of patients, which could be explained by both a decreased renal clearance of the biomarker, and increased expression in renal dysfunction, either in an acute setting or reflecting a subclinical previous kidney disease through early renal endothelial dysfunction [40,41,42]. Similar to the association between GDF-15 and AF, although age was correlated with GDF-15 pericardial fluid and serum levels (*p* = 0.006 and *p* = 0.004, respectively—data not shown), patients had a similar age independently of AKI events. Thus, even though age could be a potential explanation for the association between the biomarker and AKI, the results appear not be confounded.

### 4.7. Translational Implications

Either GDF-15 or NT-pro-BNP presented a good correlation between biological fluids, although a logarithm transformation was used. Few studies have reported identical correlation between serum and pericardial fluid [27,33], which supports comparability of the biomarkers between biological matrices. However, this manuscript suggests that pericardial fluid might have an increased clinical significance in cardiac diseases, namely, atrial fibrillation in severe aortic stenosis. In the cardiac surgery setting, pericardial fluid becomes readily available, thus becoming a useful diagnostic and prognostic tool, either for predicting postoperative complications, such as AKI, or for risk stratification of AF patients, since a worse remodeling, demonstrated here as being correlated with pericardial fluid biomarkers, is associated with increased all-cause mortality after aortic valve replacement [43]. Atrial fibrosis and remodeling is associated with recurrence after ablation, as well as stroke history and atrial thrombosis when added to other classical risk scores, such as CHA_2_DS_2_-VASc [44]. Thus, identifying patients with increased atrial fibrosis through pericardial fluid GDF-15 and NT-pro-BNP could signal patients with a higher risk of adverse outcomes, providing a better treatment adjustment and follow-up. Quantifying these biomarkers is easier and cheaper than other methods of estimating atrial remodeling (e.g., cardiac magnetic resonance). Furthermore, serum NT-pro-BNP could potentially identify patients with other forms of AF (i.e., paroxysmal) that have a negative routine EKG, prompting Holter monitoring or a tighter follow-up. Eventually, these patients could be treated to stop progression to more permanent forms of AF.

This study, nevertheless, has some limitations. The low number of patients with AF could decrease the certainty of our conclusions. On the other hand, to the best of our knowledge, this is the first study that associates both pericardial fluid and serum GDF-15 and NT-pro-BNP in AVR surgery with AF, as well as acute kidney injury. Additionally, this work sheds light on possible mechanisms, associating GDF-15 with AF pathophysiology, since it is related, to some degree, to atrial matrix remodeling.

## 5. Conclusions

AF in severe aortic stenosis is associated with increased levels of pericardial fluid GDF-15. Pericardial fluid NT-pro-BNP had a higher diagnostic accuracy for AF than its serum counterpart. AKI after AVR surgery is predicted by GDF-15 in both biological matrices. Pericardial fluid but not serum NT-pro-BNP and GDF-15 are markers of atrial matrix remodeling.

## Figures and Tables

**Figure 1 diagnostics-11-01422-f001:**
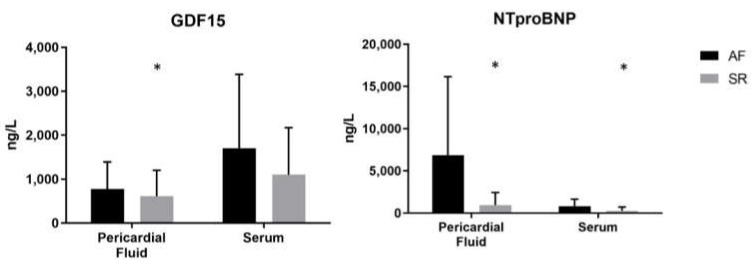
Serum and pericardial fluid GDF-15 and NT-pro-BNP quantifications between atrial fibrillation and sinus rhythm patients. Sample number: pericardial fluid GDF-15 (SR = 48; AF = 17); serum GDF-15 (SR = 86; AF = 9); pericardial fluid NT-pro-BNP (SR = 47; AF = 17); serum NT-pro-BNP (SR = 88; AF = 9). Mann–Whitney U test was used for all comparisons; * *p* < 0.05.

**Figure 2 diagnostics-11-01422-f002:**
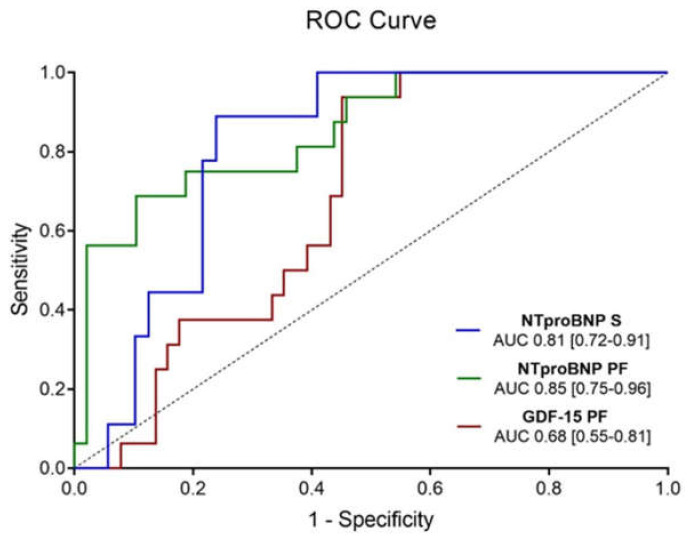
AUROC Curves of pericardial fluid and serum NT-pro-BNP and pericardial fluid GDF-15.

**Figure 3 diagnostics-11-01422-f003:**
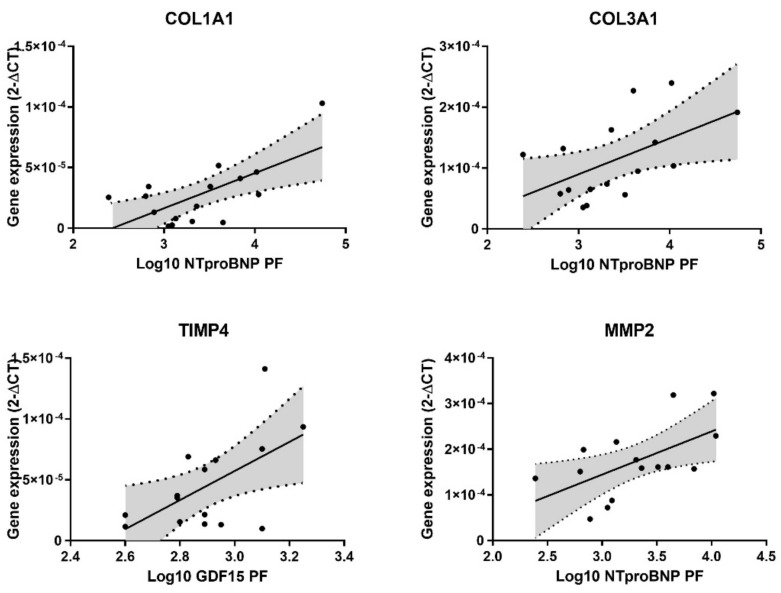
Correlation between pericardial fluid GDF-15 and NT-pro-BNP and atrial extracellular matrix gene expression.

**Figure 4 diagnostics-11-01422-f004:**
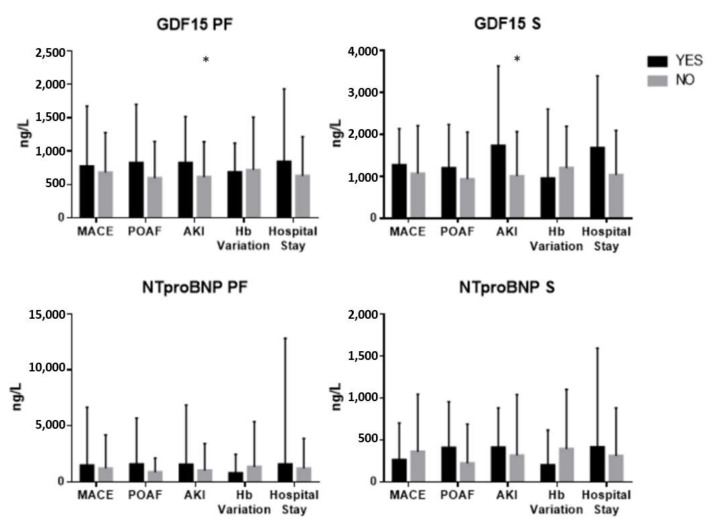
Differences between pericardial fluid and serum NT-pro-BNP and GDF-15 and 30-day postoperative outcomes. Mann–Whitney U test was used for all comparisons; * *p* < 0.05. Sample number: Pericardial fluid GDF-15 (MACE: Y = 11, *n* = 54; AKI: Y = 21, *n* = 44; POAF: Y = 10, *n* = 39; Hb Variation: Y = 17, *n* = 47; Hospital Stay: Y = 14, *n* = 51); Serum GDF-15 (MACE: Y = 19, *n* = 76; AKI: Y = 22, *n* = 73; POAF: Y = 29, *n* = 57; Hb Variation: Y = 20, *n* = 74; Hospital Stay: Y = 21, *n* = 74); Pericardial fluid NT-pro-BNP (MACE: Y = 10, *n* = 54; AKI: Y = 21, *n* = 43; POAF: Y = 10, *n* = 38; Hb Variation: Y = 17, *n* = 46; Hospital Stay: Y = 14, *n* = 50); Serum NT-pro-BNP (MACE: Y = 19, *n* = 78; AKI: Y = 22, *n* = 75; POAF: Y = 29, *n* = 59; Hb Variation: Y = 20, *n* = 76; Hospital Stay: Y = 21, *n* = 76).

**Figure 5 diagnostics-11-01422-f005:**
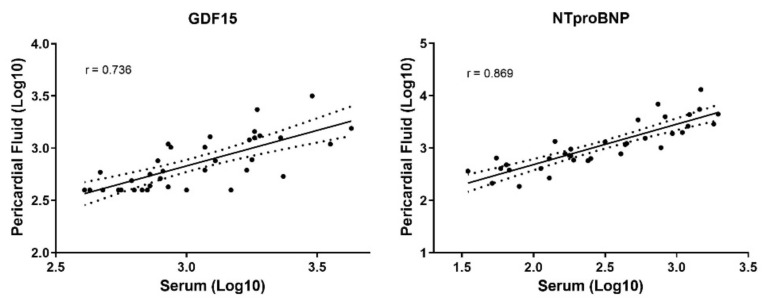
Correlation between serum and pericardial fluid quantifications of both GDF-15 and NT-pro-BNP biomarkers. Measure of association—–Pearson’s correlation coefficient.

**Table 1 diagnostics-11-01422-t001:** Baseline patient characteristics.

Variable [%(*n*)]	Total (*n* = 126)	AF (*n* = 19)	SR (*n* = 107)	*p* Value
Age, years [mean ± sd]	72.2 ± 8.4	74.0 ± 9.1	71.9 ± 8.3	0.306
Gender				
Male	48.4 (61)	42.1 (8)	49.5 (53)	0.551
Dyslipidemia	66.7 (84)	73.7 (14)	67.3 (70)	0.583
Hypertension	75.4 (95)	72.2 (13)	78.8 (82)	0.532
Diabetes Mellitus	31.7 (40)	26.3 (5)	33.3 (35)	0.547
Chronic Lung Disease	26.2 (33)	26.3 (5)	26.4 (28)	0.993
Smoking	7.1 (9)	0.0 (0)	8.9 (9)	0.352
PAD	0.8 (1)	0.0 (0)	1.0 (1)	1.000
CVD	10.3 (13)	11.1 (2)	11.6 (11)	1.000
Previous AMI	3.2 (4)	0.0 (0)	4.1 (4)	1.000
CAD	18.3 (23)	10.5 (2)	21.2 (21)	0.360
NYHA ≥ II	64.3 (81)	75.0 (12)	78.4 (69)	0.749
Angina	17.5 (22)	3.0 (20)	19.0 (22.4)	1.000
Syncope/Lipothymia	8.7 (11)	0.0 (0)	11.0 (12.9)	0.209
Basal Creatinine, mg/dL [median (Q1-Q3)]	0.8 (0.7–1.0)	0.9 (0.8–1.0)	0.8 (0.6–1.0)	0.165
Basal Hemoglobin, g/dL [mean ± sd]	13.2 ± 1.4	13.1 ± 0.8	13.2 ± 1.5	0.912
	Echocardiography			
Ejection Fraction, % [mean ± sd]	61.1 ± 7.1	58.6 ± 2.7	61.6 ± 7.6	0.305
LAD, mm [mean ± sd]	42.8 ± 9.7	49.4 ± 7.2	41.0 ± 9.6	**0.038**
LVED, mm [mean ± sd]	49.6 ± 5.4	52.0 ± 6.6	49.0 ± 5.0	0.196
IVS, mm [mean ± sd]	13.4 ± 2.4	12.9 ± 3.4	13.6 ± 2.2	0.505
LVPW, mm [mean ± sd]	11.3 ± 1.7	11.5 ± 1.8	11.2 ± 1.7	0.736
AMG, mmHg [mean ± sd]	82.1 ± 21.2	78.4 ± 19.7	82.9 ± 21.6	0.444
AMeanG, mmHg [mean ± sd]	50.4 ± 12.6	48.8 ± 12.7	50.8 ± 12.6	0.559
AVA, cm^2^ [mean ± sd]	0.8 ± 0.2	0.7 ± 0.2	0.8 ± 0.2	0.576

Baseline patient demographics and echocardiographic parameters by heart rhythm status. PAD—Peripheral Artery Disease; CVD—Cerebrovascular Disease; CAD—Coronary Artery Disease; NYHA—New York Heart Association; LAD—Left Atrium Diameter; LVED—Left Ventricle End-diastolic Diameter; IVS—Interventricular Septum; LVPW—Left Ventricle Posterior Wall; AMG—Aortic Maximum Gradient; AMeanG—Aortic Mean Gradient; AVA—Aortic Valve Area.

**Table 2 diagnostics-11-01422-t002:** Correlations between biomarkers and atrial matrix remodeling.

Variable	PF GDF-15	S GDF-15	PF NT-Pro-BNP	S NT-Pro-BNP
Fibrosis	0.001 [−0.119–0.121], *p* = 0.983	−0.037 [−0.180–0.106], *p* = 0.579	0.007 [−0.046–0.059], *p* = 0.794	0.011 [−0.062–0.085], *p* = 0.741
COL1A1	2.1 × 10^−5^ [−6.2 × 10^−5^–1.0 × 10^−4^], *p* = 0.601	3.5 × 10^−5^ [−3.3 × 10^−5^–1.0 × 10^−4^], *p* = 0.226	2.9 × 10^−5^ [1.0 × 10^−5^–4.8 × 10^−5^], *p* = 0.005	4.0 × 10^−6^ [−4.9 × 10^−5^–5.7 × 10^−5^], *p* = 0.845
COL3A1	−1.6 × 10^−5^ [−2.3 × 10^−4^–1.9 × 10^−4^], *p* = 0.873	7.5 × 10^−5^ [−2.2 ×10^−4^–3.7 × 10^−4^], *p* = 0.516	6.0 × 10^−5^ [6.0 × 10^−6^–1.1 × 10^−4^], *p* = 0.033	5.4 × 10^−5^ [−1.3 × 10^−4^–2.4 × 10^−4^], *p* = 0.460
TIMP4	1.2 × 10^−4^ [1.8 × 10^−5^–2.2 × 10^−4^], *p* = **0.025**	1.1 × 10^−5^ [−1.7 × 10^−4^–1.9 × 10^−4^], *p* = 0.861	−9.4 × 10^−6^ [1.2 × 10^−4^–9.9 × 10^−5^], *p* = 0.801	1.8 × 10^−5^ [−1.8 × 10^−5^–5.4 × 10^−5^], *p* = 0.309
TIMP1	−0.001 [−0.002–0.001], *p* = 0.339	7.0 × 10^−5^ [−0.003–0.004], *p* = 0.959	7.8 × 10^−5^ [-4.2 × 10^−4^–0.001], *p* = 0.739	0.001 [-4.1 × 10^−4^–0.003], *p* = 0.108
MMP2	−5.5 × 10^−5^ [−3.0 × 10^−4^–1.9 × 10^−4^], *p* = 0.632	-9.2 × 10^−5^ [-0.001–3.3 × 10^−4^], *p* = 0.574	9.5 × 10^−5^ [1.6 × 10^−5^–1.7 × 10^−4^], *p* = 0.023	9.3 × 10^−5^ [-1.6 × 10^−4^–3.4 × 10^−4^], *p* = 0.359
MMP9	−2.2 × 10^−6^ [−2.9 × 10^−5^–2.4 × 10^−5^], *p* = 0.860	−3.7 × 10^−5^ [−1.1 × 10^−4^–4.1 × 10^−5^], *p* = 0.257	9.0 × 10^−8^ [−1.0 × 10^−6^–2.0 × 10^−6^], *p* = 0.900	1.7 × 10^−5^ [−7.2 × 10^−5^–3.9 × 10^−5^], *p* = 0.446

Correlation between pericardial fluid/serum NT-pro-BNP and GDF-15 and atrial matrix remodeling. Results are presented as β coefficients and respective 95% Confidence Intervals. PF—Pericardial Fluid. S—Serum.

**Table 3 diagnostics-11-01422-t003:** Postoperative outcomes at 30 days of follow-up.

Variable [%(*n*)]	AF	SR	*p* Value
MACE	15.8 (3)	19.6 (21)	1.000
AKI	57.9 (11)	21.5 (23)	**0.001**
Hb Variation, g/dL [mean ± sd]	4.6 ± 1.1	3.9 ± 1.6	**0.017**
Hospital Stay (>75th Percentile)	9.0 (3)	7.0 (5)	0.053

30-day postoperative outcomes between atrial fibrillation and sinus rhythm patients. MACE—Major Adverse Cardiovascular Events; AKI—Acute Kidney Injury; Hb—Hemoglobin.

**Table 4 diagnostics-11-01422-t004:** Impact of target biomarkers in 30-day postoperative outcomes.

Outcome [Median (Q1–Q3)]		PF GDF-15	*p* Value	S GDF-15	*p* Value	PF NT-Pro-BNP	*p* Value	S NT-Pro-BNP	*p* Value
MACE	Y	714.1 (400.0–1009.8)	0.760	1268.0 (807.7–1675.0)	0.602	1479.4 (593.8–5782.5)	0.824	260.1 (159.8–602.8)	0.856
	*n*	703.5 (476.2–1044.3)		1069.0 (769.3–1910.0)		1201.5 (633.1–3625.5)		358.6 (155.2–840.1)	
AKI	Y	826.5 (622.9–1316.0)	**0.027**	1728.5 (963.2–2863.0)	**0.015**	1554.0 (864.5–6178.5)	0.096	409.7 (171.4–643.7)	0.654
	*n*	621.2 (406.7–912.0)		1007.0 (717.9–1777.0)		1014.0 (476.6–2902.0)		316.6 (155.8–878.7)	
POAF	Y	826.5 (419.3–1230.5)	0.435	1194.0 (902.5–1942.5)	0.076	1578.5 (586.4–4712.8)	0.298	407.4 (199.1–745.6)	0.096
	*n*	606.6 (400.0–962.0)		935.2 (700.5–1821.0)		864.6 (460.2–1729.8)		221.7 (129.0–596.3)	
Hb Variation (>75th Percentile)	Y	686.7 (403.4–838.5)	0.291	954.2 (692.4–2337.5)	0.530	784.4 (408.6–2090.5)	0.146	199.1 (104.8–522.6)	0.182
	*n*	733.2 (494.0–1205.8)		1203.0 (856.9–1849.3)		1342 (664.5–4712.8)		392.5 (166.8–876.0)	
Hospital Stay (>75th Percentile)	Y	826.5 (467.3–1287.5)	0.348	1675.0 (860.8–2574.0)	0.085	1579.5 (633.1–11867.3)	0.314	412.0 (151.8–1331.5)	0.386
	*n*	654.3 (437.4–1024)		1035.5 (760.0–1820.5)		1201.5 (596.0–3279.8)		310.3 (156.8–727.5)	

30-day postoperative outcomes and their association with pericardial fluid/serum GDF-15 and NT-pro-BNP. MACE—Major Adverse Cardiovascular Events; AKI—Acute Kidney Injury; Hb—Hemoglobin. PF—Pericardial Fluid. S—Serum.

## Data Availability

The data presented in this study are available on request from the corresponding author. The data are not publicly available due to ethical restrictions.

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
