# Peer review of "Pericardial NT-Pro-BNP and GDF-15 as Biomarkers of Atrial Fibrillation and Atrial Matrix Remodeling in Aortic Stenosis"

_diagnostics, 2021, doi:10.3390/diagnostics11081422_

Round 1
Reviewer 1 Report
Reviewing the manuscript entitled, “Pericardial NT-pro-BNP and GDF-15 as biomarkers of atrial fibrillation and atrial matrix remodeling in aortic stenosis” by Fragão-Marques M. et al., this is an article focusing on pericardial NT-pro-BNP and GDF-15 as biomarkers of atrial fibrillation post AVR patients using a posthoc analysis. Although pericardial NT-pro-BNP and GDF-15, biomarkers that can only be used under extremely limited conditions after surgery, the relevance with AKI is extremely interesting.
Concerns
The average age of the subjects is as old as 72.2 years. The authors need to describe the association between aging including pathology and GDF-15, and explain it in the discussion or statistically correct the effects on the results.
There is no heart rate in the survey items, why? Heart rate is one of the important factors that determine the prognosis of heart failure. Especially in the case of atrial fibrillation, heart rate control does not improve prognosis. Therefore, I think GDF- 15 and heart rate are an interesting linkage. The authors need to describe it in the discussion.
In Figure 3, the authors need to add negative data such as NT-proBNP or GDF-15 in serum.
Author Response
Response to Reviewers
Thank you for giving us the opportunity to submit a revised draft of the manuscript “Pericardial NT-pro-BNP and GDF-15 as biomarkers of atrial fibrillation and atrial matrix remodeling in aortic stenosis” for publication in the journal Diagnostics. We appreciate the time and effort that you and the reviewers dedicated to providing feedback on our manuscript and are grateful for the insightful comments on and valuable improvements to our paper. We have incorporated most of the suggestions made by the reviewers. Those changes are highlighted within the manuscript.
Reviewer 1
The authors have now discussed the association between GDF-15 and age, either regarding AF patients or the occurrence of AKI. An association between age and pericardial fluid and serum GDF-15 values was performed (p values in the discussion). New paragraphs were added to the discussion (pages 16-17, lines 347-353; pages 19-20, lines 419-424). Moreover, a paragraph regarding the possible relationship between GDF-15 and heart rate was added in page 17, lines 353-356.

Reviewer 2 Report
The study by Fragao-Marques et al. aimed at evaluating the association between pericardial fluid and serum GDF-15 and NT-pro-BNP and atrial fibrillation in patients with severe aortic stenosis undergoing surgical AVR, their association with atrial matrix remodeling, and their predictive value in postoperative outcomes. The main findings are the following: pericardial fluid GDF-15 and NT-pro-BNP are increased in patients with AF and severe aortic stenosis undergoing AVR, and they correlate with atrial matrix remodeling. Moreover, both pericardial fluid and serum GDF-15 are predictors of postoperative AKI and bleeding. In order to make these findings more interesting to the readers it would be useful to expand the paragraph "translational indications", describing how they can really change clinical practice and influence medical decisions. At the same time, I would shorten and summarize the part dedicated to the results, mainly basing it on tables and graphs.Author Response
Response to Reviewers
Thank you for giving us the opportunity to submit a revised draft of the manuscript “Pericardial NT-pro-BNP and GDF-15 as biomarkers of atrial fibrillation and atrial matrix remodeling in aortic stenosis” for publication in the journal Diagnostics. We appreciate the time and effort that you and the reviewers dedicated to providing feedback on our manuscript and are grateful for the insightful comments on and valuable improvements to our paper. We have incorporated most of the suggestions made by the reviewers. Those changes are highlighted within the manuscript.
Reviewer 2
Thank you for your dedication in reviewing this manuscript. The authors agree with the relevance in expanding the paragraph about the translational implications of this work. The appropriate changes were made to the manuscript (page 20, lines 436-445).
Furthermore, the authors have shortened the results throughout the manuscript, including the following headlines: baseline echocardiographic assessment (page 11); atrial matrix remodeling (page 13); 30-day postoperative outcomes (atrial fibrillation, page 13; and pericardial fluid and serum GDF-15 and NT-pro-BNP, page 14).
